# OpenReview forum: "Beyond Sharp Minima: Robust LLM Unlearning via Feedback-Guided Multi-Point Optimization"
_ICLR.cc/2026/Conference — ICLR 2026 Conference Withdrawn Submission_

### Official Review · Reviewer_ygDs · 2025-10-24

**Soundness:** 2
**Presentation:** 3
**Contribution:** 2
**Rating:** 2
**Confidence:** 5

**Summary:**

The paper investigates the robustness of large language model (LLM) unlearning against relearning and jailbreak attacks. It identifies that existing unlearning methods drive parameters toward sharp minima, leaving models vulnerable to rapid recovery of forgotten knowledge through minimal fine-tuning. To address this, the authors propose StableUN, a feedback-guided bi-level optimization framework that introduces two complementary mechanisms: forgetting feedback, which applies adversarial perturbations to ensure stable forgetting, and remembering feedback, which preserves model utility. These feedback signals are integrated via gradient harmonization to balance robustness and utility. Experiments on WMDP and MUSE benchmarks show that StableUN improves resistance to relearning and jailbreak attacks while maintaining comparable performance on general tasks.

**Strengths:**

1. The figures and tables are clear and easy to interpret.
2. The formulation is well specified and consistent throughout.
3. The experiments cover both WMDP and MUSE benchmarks, which strengthens the empirical support.

**Weaknesses:**

1. The paper claims two main contributions. The first is the use of a bi-level optimization framework, which has already been explored in prior work [1]. The second is adding perturbations to the forgetting gradient, but this idea has also been proposed in earlier studies [2-3]. The authors do not clearly explain how their method differs from these existing approaches.

2. The paper lacks many important baselines. Recent robust unlearning methods [2-4] have not been included for comparison.

3. Using only MMLU to measure model utility is insufficient, as the observed robustness improvement might come at the cost of degraded utility in other aspects. It is recommended to include additional utility evaluations on WMDP for a more comprehensive assessment [4].

4. Computational overhead is not quantified. The bi-level procedure appears to hold two sets of gradients and adds extra perturbation evaluations, which likely increases both wall-clock time and peak memory versus standard unlearning. Please report a compute and memory comparison under matched settings (same model, context length, batch size, optimizer, GPUs).

> [1] Reisizadeh, Hadi, et al. "BLUR: A Bi-Level Optimization Approach for LLM Unlearning." arXiv preprint arXiv:2506.08164 (2025).
>
> [2] Sheshadri, Abhay, et al. "Latent adversarial training improves robustness to persistent harmful behaviors in llms." arXiv preprint arXiv:2407.15549 (2024).
>
> [3] Fan, Chongyu, et al. "Towards llm unlearning resilient to relearning attacks: A sharpness-aware minimization perspective and beyond." arXiv preprint arXiv:2502.05374 (2025).
>
> [4] Tamirisa R, Bharathi B, Phan L, et al. Tamper-resistant safeguards for open-weight llms[J]. arXiv preprint arXiv:2408.00761, 2024.
>
> [5] Che Z, Casper S, Kirk R, et al. Model tampering attacks enable more rigorous evaluations of llm capabilities[J]. arXiv preprint arXiv:2502.05209, 2025.

**Questions:**

See weakness.

---

### Official Review · Reviewer_vhGS · 2025-10-31

**Soundness:** 3
**Presentation:** 3
**Contribution:** 2
**Rating:** 2
**Confidence:** 4

**Summary:**

This paper introduces StableUN, a robust unlearning framework for large language models that mitigates the problem of sharp minima in loss landscapes. The authors observe that existing unlearning methods easily fall into unstable optima, making forgotten knowledge recoverable through relearning attacks. To address this, StableUN incorporates multi-point neighborhood optimization and gradient harmonization to balance forgetting and retention. Experiments on multiple benchmarks demonstrate improved robustness and stability without sacrificing model utility.

**Strengths:**

1.	The paper provides a novel perspective on the relearning attack problem by identifying the overlooked issue of sharp minima in existing unlearning optimization methods. The proposed dual-feedback mechanism effectively mitigates this issue, as demonstrated by the experimental results.
2.	The paper is well written and clearly organized, making the technical content easy to fellow. The use of illustrative figures and algorithmic descriptions helps clarify the core ideas and enhances overall readability.

**Weaknesses:**

1.	The motivation of the paper could be further strengthened. Although the authors identify and empirically verify the gradient phenomenon in GA, the current motivation is somewhat narrow. It would be more convincing to include similar observations from other unlearning paradigms to comprehensively illustrate the sharp basin issue. Additionally, referencing prior analyses on relearning attacks and their underlying causes would help contextualize the problem.
2.	The paper lacks sufficient comparison with existing relearning-attack mitigation methods. It would be valuable to include quantitative and qualitative comparisons with recent unlearning repair frameworks such as [1], rather than only demonstrating that the proposed method is effective. A more detailed comparison would also better highlight the distinctiveness and originality, particularly since [1] also adopts sharpness-aware optimization strategies.
3.	The gradient clipping strategy proposed to resolve gradient conflicts is not entirely new within the machine unlearning. The authors should cite related works, such as GDR-GMA[2], to better position this contribution within existing research.
4.	The paper also lacks analysis of computational overhead. A discussion of the additional cost introduced by the proposed robustness enhancements is necessary.
5.	Experiments on newer and larger LLMs (e.g., LLaMA-3) would further demonstrate the generality of the framework.
[1] Chongyu Fan, Jinghan Jia, Yihua Zhang, Anil Ramakrishna, Mingyi Hong, and Sijia Liu. Towards llm unlearning resilient to relearning attacks: A sharpness-aware minimization perspective and beyond. In Forty-second International Conference on Machine Learning, 2025.
[2] Lin, Shen, et al. "GDR-GMA: Machine Unlearning via Direction-Rectified and Magnitude-Adjusted Gradients." Proceedings of the 32nd ACM International Conference on Multimedia. 2024.

**Questions:**

Please review the questions outlined in the cons above.
Additionally, I find it difficult to understand the motivation and working principle of the bi-level optimization mentioned in the paper. It seems that the discussion suddenly shifts from the two feedback mechanisms to bi-level optimization. What is the significance of introducing this structure? The paper also states that it is inspired by meta-learning. What kind of inspiration does meta-learning provide? In what way are they similar, and what is the intended purpose of this design? The description and figures related to this part are also somewhat unclear to me. As I understand it, the inner loop seems to generate the unlearning gradients for the subsequent feedback without any optimization, but I am not sure if my understanding is correct. In the figure, both loops are shown at the bottom, and do the two colored lines represent these two loops? I hope the authors can improve the figure and corresponding explanations to make it easier to understand.
Moreover, although there have been studies on relearning attacks, I wonder whether the phenomenon might occur because some similar samples remain in the retained dataset after certain data are deleted, making the forgotten samples easier to relearn. For instance, it may be interesting to examine how the model relearns samples that are similar to the forgotten set but were not part of the original training data. Exploring sample similarity or investigating the intrinsic difficulty of forgetting certain samples could be a valuable research direction. Alternatively, the authors could provide relevant findings and conclusions from prior studies on relearning attacks to further clarify this issue.

---

### Official Review · Reviewer_HhnP · 2025-11-01

**Soundness:** 3
**Presentation:** 3
**Contribution:** 2
**Rating:** 2
**Confidence:** 4

**Summary:**

This paper presents StableUN, a feedback-guided bi-level optimization framework for robust unlearning in large language models. The authors argue that existing unlearning methods optimize forgetting loss at single parameter points, leading to sharp minima that make models vulnerable to relearning attacks. StableUN introduces two feedback signals- forgetting feedback using adversarial perturbations to probe parameter neighborhoods, and remembering feedback to preserve utility-combined through a gradient harmonization mechanism. Experiments on WMDP and MUSE benchmarks suggest improved robustness to relearning and jailbreak attacks while maintaining model utility.

**Strengths:**

1. The paper addresses a relevant problem in LLM unlearning, focusing on robustness against adversarial recovery.

2. The idea of connecting sharp minima with vulnerability to relearning provides an intuitive and optimization-oriented perspective on unlearning reliability.

3. The proposed method is modular and can be integrated with existing gradient-based unlearning approaches.

4. The empirical study covers multiple models (Zephyr-7B, Mistral-7B, LLaMA2-7B, ICLM-7B) and datasets (WMDP-bio/cyber, MUSE News/Books).

5. The paper is generally clear and well-organized, and visualizations help convey the core idea effectively.

**Weaknesses:**

1.The central claim that sharp minima cause unlearning vulnerability lacks rigorous theoretical or quantitative support. The argument is largely heuristic, relying on visualizations (Fig. 2, p4, l180-191) rather than formal curvature analysis or stability measures. No formal curvature or analysis accompanies the claim that “single point optimization driving parameters toward sharp, unstable minima” (p2, l76-78).

2.The proposed framework adds substantial algorithmic complexity with multiple feedback loops, perturbations, and gradient projections, yet the observed gains are modest. The paper does not discuss computational overhead, scalability, or efficiency trade-offs.

3.The novelty is limited. The method builds heavily on existing ideas such as sharpness-aware minimization, adversarial perturbation training, and gradient projection from multi-task learning. The overall framework feels more like an engineered combination of known components than a conceptually new algorithm.

4.The empirical evaluation is comparing only to older baselines like GA, GA+GD, and NPO instead of modern baselines.

5.The experiments focus exclusively on moderate-scale benchmarks and models. There is no evidence that the method generalizes to large instruction-tuned or multimodal LLMs where optimization dynamics differ substantially.

6.The analysis of forgetting-remembering trade-offs is shallow. While figures indicate only minor utility degradation, the paper offers no quantitative exploration of how to balance these competing objectives. It is possible that robustness gains simply arise from excessive smoothing that reduces generalization capacity. Although Appendix A.6 provides a sensitivity test on the perturbation radius ρ and feedback weights $λ_f$, $λ_r$, it lacks the discussion of how these affect the balance between robustness and utility.

7.The claim that the approach enhances AI safety and trustworthy unlearning is overstated. The datasets used (WMDP, MUSE) are synthetic or sanitized, and no experiments demonstrate mitigation of real-world data leakage.

8.The paper provides little mechanistic insight into why the proposed bi-level feedback structure improves stability beyond the intuition of flatter minima. No curvature metrics, Hessian spectra, or gradient sensitivity analyses are presented to substantiate the claim anywhere in Sec. 4 or Appendix A.

9.The feasibility remains uncertain. The need to compute multiple perturbations per iteration and harmonize gradients will likely increase training time substantially. The authors do not report runtime or GPU-hour comparisons against baselines.

**Questions:**

Please refer to the weaknesses.

---

### Official Review · Reviewer_8mWd · 2025-11-01

**Soundness:** 2
**Presentation:** 3
**Contribution:** 3
**Rating:** 2
**Confidence:** 4

**Summary:**

This paper tackles the fragility of LLM unlearning to small-shot relearning. The authors trace the issue to unstable neighborhoods in parameter space and build a feedback-guided, bi-level procedure that probes local neighborhoods, extracts two feedback signals for forgetting and retaining, and then harmonizes their directions with a projection step before updating. The presentation walks the reader through the neighborhood idea, the feedback construction, and the final update schedule, and illustrates the intended geometry with slice visualizations and a block diagram of the learning loop.

**Strengths:**

1.	The paper isolates a concrete failure mode and builds a method that is explicitly designed around neighborhood behavior, not just pointwise losses.

2.	The feedback mechanism is clearly specified, and the algorithmic flow from temporary parameters to a final update can be followed step by step.

**Weaknesses:**

1. The manuscript says sharp neighborhoods explain why small-shot relearning works, but it does not actually connect the stated objective in Eq. 4 to a measurable recovery tendency. Eq. 4 minimizes a sharpness-augmented target, yet the paper does not report random-direction curvature, local Lipschitz, or similar neighborhood metrics that would link this target to reversibility.
2. The declared scalar objective and the realized update do not match. Eq. 11 embeds half of the forgetting loss in each branch and Eq. 13 sums the two branches, while Eq. 10 contains the forgetting loss once; this double-counts its gradient and skews the step away from the nominal objective, creating a systematic bias in the descent direction.
3. The projection in Eq. 12 divides by the squared norm of the retaining gradient. When that norm is small, the denominator makes the update numerically fragile and amplifies noise in flat regions, and the formulation does not include any conditioning guard for this case.
4. All flatness and sharpness are judged in weight space, but the model is adapted with LoRA. The same function can be written as many different factor pairs (A,B), and simple rescalings of those factors change gradient and curvature magnitudes while the function stays the same.

**Questions:**

Please refer to the weakness above.

---

### Note · Authors · 2026-01-17

I have read and agree with the venue's withdrawal policy on behalf of myself and my co-authors.